

# Stable dominance of parasitic dinoflagellates in Antarctic sponges

Marileyxis R. López-Rodríguez[1], Catherine Gérikas Ribeiro[1], Susana Rodríguez-Marconi[1], Génesis Parada-Pozo[1,2], Maria Manrique-de-la-Cuba[1] and Nicole Trefault[1,2,3]

[1] GEMA Center for Genomics, Ecology & Environment, Universidad Mayor, Santiago, Chile
[2] Millenium Nucleus in Marine Agronomy of Seaweed Holobionts (MASH), Puerto Montt, Chile
[3] FONDAP Center IDEAL- Dynamics of High Latitude Marine Ecosystem, Valdivia, Chile

Corresponding author
Nicole Trefault,
nicole.trefault@umayor.cl

## ABSTRACT

**Background:** Marine sponges are dominant components of Antarctic benthos and representative of the high endemism that characterizes this environment. All microbial groups are part of the Antarctic sponge holobionts, but microbial eukaryotes have been studied less, and their symbiotic role still needs to be better understood. Here, we characterize the dynamics of microbial eukaryotes associated with Antarctic sponges, focusing on dinoflagellates over three summer periods to better understand the members, interannual variations, and trophic and lifestyle strategies.

**Results:** The analysis revealed that dinoflagellates dominate microeukaryotic communities in Antarctic sponges. The results also showed significant differences in the diversity and composition of dinoflagellate communities associated with sponges compared to those in seawater. Antarctic sponges were dominated by a single dinoflagellate family, Syndiniales Dino-Group-I-Clade 1, which was present in high abundance in Antarctic sponges compared to seawater communities. Despite minor differences, the top microeukaryotic amplicon sequence variants (ASVs) showed no significant interannual abundance changes, indicating general temporal stability within the studied sponge species. Our findings highlight the abundance and importance of parasitic groups, particularly the classes Coccidiomorphea, Gregarinomorphea, and Ichthyosporea, with the exclusive dominance of Syndiniales Dino-Group-I-Clade 1 within sponges.

**Conclusions:** The present study comprehensively characterizes the microbial eukaryotes associated with Antarctic sponges, showing a remarkable stability of parasitic dinoflagellates in Antarctic sponges. These findings underscore the significant role of parasites in these marine hosts, with implications for population dynamics of the microeukaryome and the holobiont response to a changing ocean.

## INTRODUCTION

Marine sponges are fundamental and dominant components of the benthos in Antarctica (*Costa et al., 2023*). Through filter-feeding nutrition, these primitive animals play a crucial

role in marine environments. They act as ecosystem engineers, providing habitat and food for other organisms while critically involved in nutrient cycling and biogeochemical processes (*McClintock et al., 2005*; *Bell, 2008*). Sponge diversity in Antarctica is lower than in tropical regions but higher than observed in temperate environments (*Vargas et al., 2015*). Due to the isolation of the Antarctic continent by the Polar Front, sponges have high levels of endemism in this continent (*Downey et al., 2012*; *Kersken, Feldmeyer & Janussen, 2016*). The considerable spatial heterogeneity that Antarctic sponges provide for the colonization of epibionts and the eminently planktivorous diets in an environment marked by fluctuations in seasonal phytoplankton blooms also characterized these unique and primitive animals (*Cattaneo-Vietti et al., 2000*; *McClintock et al., 2005*).

Sponges are also known for associating with a diverse range of microorganisms, forming an ecological unit referred to as a sponge holobiont. In this unit, the assemblage of sponge-associated microorganisms is responsible for various essential functions for the survival and ecological success of the host (*Webster & Thomas, 2016*; *Pita et al., 2018*). These symbiotic communities engage in various biological interactions, turning the sponge holobiont into a dynamic system (*Pita et al., 2018*). Sponge-associated microorganisms include viruses, bacteria, archaea, and microbial eukaryotes. Unfortunately, not all microbial components of the sponge holobiont have been equally studied, and there is a significant gap in the knowledge regarding sponge-associated microbial eukaryotes, which is even more pronounced in Antarctica.

The Antarctic sponge holobiont harbors a diverse community of microorganisms where Proteobacteria (mainly Alphaproteobacteria and Gammaproteobacteria) and Bacteroidota, as well as the archaeal phylum Thaumarchaeota, are the most abundant non-eukaryotic groups (*Webster et al., 2004*; *Rodríguez-Marconi et al., 2015*; *Cárdenas et al., 2019*; *Moreno-Pino et al., 2020*; *Ruocco et al., 2021*). Previous studies observed that the eukaryotic community is dominated by diatoms such as Fragilariopsis and Thalassiosira; a diversity of fungi, including Geomyces and Penicillium, and other unicellular organisms like dinoflagellates, cryptophytes, and chlorophytes (*Cerrano et al., 2004*; *Henríquez et al., 2014*; *Rodríguez-Marconi et al., 2015*; *Moreno-Pino et al., 2020*). These microbial communities exhibit a different taxonomic composition than the microorganisms from the surrounding waters (*Rodríguez-Marconi et al., 2015*; *Sacristán-Soriano, Pérez Criado & Avila, 2020*; *Moreno-Pino et al., 2020*; *Cristi et al., 2022*). Overall, microbial eukaryotes associated with Antarctic sponges show greater diversity than planktonic eukaryotic communities (*Rodríguez-Marconi et al., 2015*).

The unique metabolic potential of the Antarctic sponge microbiome contributes to the survival and growth of the holobiont in this challenging environment (*Moreno-Pino et al., 2020*, *2024*). The roles of these symbiotic communities include nutrient cycling (carbon, nitrogen, phosphorus, and sulfur), biosynthesis of secondary metabolites with defensive properties, and biodegradation of xenobiotics, epibionts, food, and parasites (*Bavestrello et al., 2000*; *Cerrano et al., 2004*; *Totti et al., 2005*; *Henríquez et al., 2014*; *Steinert et al., 2019*; *Moreno-Pino et al., 2020*; *Cristi et al., 2022*; *Moreno-Pino et al., 2024*). However, to the best of our knowledge, the temporal patterns of the microeukaryome associated with the Antarctic sponges have not been previously studied.

Despite the abundance and diversity of sponge-associated dinoflagellates, little is known about this group of symbionts. Furthermore, their diversity of trophic modes and symbiotic relationships have not been deeply studied in Antarctic sponges. To date, the orders of dinoflagellates Gymnodiniales, Suessiales, and Syndiniales Group I have been identified as associated with sponges from tropical, temperate, and polar environments (*Webster et al., 2004*; *Annenkova, Lavrov & Belikov, 2011*; *He et al., 2014*; *Rodríguez-Marconi et al., 2015*; *Moreno-Pino et al., 2020*). The most extensively documented association occurs between tropical sponges of the *Cliona* genus and *Symbiodinium* (*Schönberg & Loh, 2005*; *Hill et al., 2011*), with evidence of vertical symbiont transmission across generations (*Mariani, Uriz & Turon, 2000*). These symbiotic dinoflagellates contribute not only with glucose, fatty acids, and glycerol through photosynthesis but also provide nitrogen in the form of amino acids *via* ammonia assimilation, thereby enhancing the bioerosion rates of the sponges (*Achlatis et al., 2018*, *2019*, *2021*). Furthermore, dinoflagellates produce secondary metabolites that protect the sponges against environmental stressors (*Müller et al., 2007*). These functional roles could have a significant impact in the context of the Antarctic environment.

Photosynthetic and mixotrophic dinoflagellates of the Gymnodiniales were initially described in Antarctic sponges, suggesting that these symbionts are a source of organic carbon for their hosts (*Webster et al., 2004*). More recently, a high-throughput sequencing approach revealed that Syndiniales from Group I, alveolate parasites, are the most abundant order within the microbial eukaryotes in eight Antarctic sponges, with an abundance of up to 50% (*Rodríguez-Marconi et al., 2015*). The Syndiniales comprise at least five major marine alveolate (MALV) groups and are widely distributed, abundant, and infect many hosts (*Guillou et al., 2008*). Furthermore, these organisms contribute to biogeochemical cycles by providing organic matter to the microbial loop (*Moran et al., 2022*).

In the present study, we use high-throughput sequencing of the V4 region of the 18S rRNA gene to characterize the dynamics of the microbial eukaryotes associated with Antarctic sponges. We focus on dinoflagellates, given their importance to benthic hosts and their potential key role in a changing ocean. Specifically, we aim to (i) characterize the diversity and community composition, trophic modes, and lifestyle of microbial eukaryotes, and especially dinoflagellates, associated with Antarctic sponges; and (ii) examine interannual changes in the diversity and community composition of microbial eukaryotes and especially dinoflagellates, associated with Antarctic sponges.

## MATERIALS AND METHODS

### Sample processing and collection

Adult sponge and seawater (SW) samples were collected from Chile Bay, Greenwich Island, Western Antarctic Peninsula (WAP) at depths between 8–15 m. The collection was performed during November 2019, December 2019, February 2020, March 2020, December 2020, and December 2021. Forty-seven sponge samples from the genera *Dendrilla, Mycale, Myxilla, Iophon, Isodictya*, and *Tedania*, and an undetermined Demospongiae were collected using SCUBA divers and stored separately with natural SW

until processing (sample details are available at https://github.com/mlopezrod/Antarctic-sponge_dino/tree/main/raw_data). These sponge genera were selected due to their abundance and coverage on the rocky walls from Chile Bay. Sampling scheme considered marking sampled individuals to avoid resampling them the following year, preventing tissue damage and potential changes in their microbiome. Surrounding SW was collected using a 5 L Niskin bottle and pre-filtered onboard through a 200 μm pore mesh to remove large particles. The collected SW samples were stored in acid-washed containers and kept in the dark until processing in the laboratory.

The sponge samples were rinsed with filtered SW, cleaned under a stereomicroscope to remove dirt and ectoparasites, and stored at −80 °C until processing. The separation of the microbial community intimately associated with the sponge from the host tissue was carried out following the protocol described by *Rodríguez-Marconi et al. (2015)*. SW samples were filtered through 20, 3, and 0.2 μm pore size filters using a Swinnex support system and a Cole Parmer 1-600 rpm peristaltic pump. Subsequently, filters were placed in a 2-ml cryovial and stored at −80 °C until DNA extraction.

The Chilean Antarctic Institute (INACH) issued grants for sampling under permits N° 1042/ 2019, 1043/ 2019, 802/ 2020, 803/ 2020, and 656/ 2021.

### Genomic DNA extraction, PCR amplification and sequencing

Metagenomic DNA was extracted from sponge and SW samples using the PowerSoil DNA isolation kit (Mobio) and a standard protocol with cetyltrimethylammonium bromide (CTAB) phenol-chloroform (*Doyle & Doyle, 1987*), respectively. The gDNA integrity was verified on an agarose gel (0.8% w/v) and quantified using fluorometry with Quantifluor (Promega) and the HS dsDNA Qubit™ Kit (Invitrogen™).

The V4 region of the 18S rRNA gene, targeting the microbial eukaryotic communities, was amplified with primers E572F (CYGCGGTAATTCCAGCTC) and E1009R (AYGGTATCTRATCRTCTTYG) (*Comeau et al., 2011*). The V4-PCR consisted of a 25 μl reaction with 5X HF PCR buffer, 40 mM dNTPs, 1 μM of each primer, 2 units/μL of Phusion enzyme, and 2 μL of template DNA. Amplification conditions were as follows: initial denaturation for 30 s at 98 °C, 30 cycles of 98 °C for 10 s, 55 °C for 30 s, and 72 °C for 30 s, followed by a 4 min 30 s extension at 72 °C. The amplicons were quantified with Qubit and sequenced using the Miseq Illumina platform (300 bp paired-end) at the Integrated Microbiome Resource (Dalhousie University, Halifax, Canada). Three separate sequencing runs were performed.

### Bioinformatics analysis

To infer the amplicon sequence variants (ASVs), we used the dada2 package version 1.26.0 (*Callahan et al., 2016*) in the R environment version 4.2.2 (*R Core Team, 2022*). Primers were removed using cutadapt v4.2 (*Martin, 2011*). The reads were truncated and filtered based on quality profiles of each sequencing separately using the following parameters: truncLen = c(256,194), c(216,245), and c(262,195), truncQ = 2, maxEE = c(2,2), and maxN = 0.

The sequences were analyzed on a sequencing run base to obtain specific error models and were subsequently dereplicated and merged. The reads were combined before proceeding with chimera removal. The combined reads were then collapsed using the "collapseNoMismatch" step, and chimeras were removed with the "removeBimeraDenovo" function with the "consensus" method. The ASVs were annotated with an 80% bootstrap confidence level using the Protist Ribosomal Reference database PR$^2$ v4.14.0 (*Guillou et al., 2012*), incorporating the DINOREF database (*Mordret et al., 2018*). The annotated ASV table was filtered to remove ASVs not assigned at the Kingdom, Supergroup, Division, and Class levels and those assigned to Metazoa, Bacteria, chloroplasts, mitochondria, Embryophyceae, Florideophyceae, and Phaeophyceae. Next, the ASV abundance matrix, the taxonomic assignment matrix, and the metadata were combined using phyloseq v.1.38.0 (*McMurdie & Holmes, 2013*). Singletons and samples with less than 100 sequences and 1 ASV were removed, which resulted in 26 sponge samples and 12 SW samples for further analyses (Table S1). The sequences from different SW fractions were summed up for diversity and taxonomic composition analyses. Two subsets of data were created, one comprising the Dinoflagellata division exclusively and the other excluding these microorganisms (called here as 'non-dino community').

## Data analysis

We analyzed the eukaryotic community, separating the dinoflagellates from the rest of the microbial eukaryotes due to the elevated copy numbers of the 18S rRNA gene within this group, minimizing bias in the abundance estimations.

All analyses were conducted in R version 4.2.2 (*R Core Team, 2022*). The reads were rarified using the *rarefy_to_even_depth()* function in phyloseq, based on the sample with the lowest number of reads (E60 for the non-dino community and E36 for dinoflagellates). Observed richness and Shannon's index were calculated using the *estimate_richness()* function in phyloseq. Also, we assessed phylogenetic diversity (PD) using Faith's phylogenetic diversity index (*Faith, 1992*) with the picante package (*Kembel et al., 2010*). The *plot_richness()* and *ggplot()* functions of the phyloseq and ggplot2 packages (*Wickham, 2016*) were used for the visualization of alpha diversity data (Observed richness, Shannon, and PD). To assess the significance of the comparison of alpha diversity between sponges and SW, we conducted the Wilcoxon rank sum test (*Mann & Whitney, 1947*) for unpaired samples using the *compare_means()* function in ggpubr package (*Kassambara, 2020*). Also, to evaluate if there were significant differences among the three summers, we performed a Kruskal-Wallis test (*Kruskal & Wallis, 1952*) using the *compare_means()* function for only the sponge genus with replicates across all three summers. Then, a Wilcoxon test was conducted to determine which summers significantly differ in alpha diversity indices.

Microbial community composition was analyzed using nonmetric multidimensional scaling (nMDS) based on Bray-Curtis dissimilarities and Aitchison distances using the *metaMDS()* function within the vegan package (*Oksanen et al., 2022*). ASVs with a total relative abundance of less than 0.1% were excluded from the dataset, and Hellinger's transformation for Bray-Curtis dissimilarities or the centered log-ratio in the case of

Aitchison distances were applied to normalize abundances (*Rao, 1995*; *Legendre & Legendre, 1998*; *Aitchison, 1986*). To assess the significance of differences among habitat (sponges and SW), summers, and sponge genera, we conducted an analysis of similarities (ANOSIM) and multivariate permutational analysis of variance (PERMANOVA) (*Clarke, 1993*; *Anderson, 2001*). Before performing the PERMANOVA, we excluded sponge species that were undefined and those without replicates. We utilized the permutational multivariate analysis of dispersion (PERMDISP) (*Anderson, 2006*) to detect differences in homogeneity among the groups evaluated in the PERMANOVA. The multivariate statistical analyses used the *anosim()*, *adonis2()*, and *betadisper()* functions from the vegan package.

For taxonomic composition analysis, the number of reads from each sample was normalized by the median sequencing depth. The absolute abundance of ASVs was transformed into relative abundance using the *transform_sample_counts()* function in phyloseq. To preprocess the data, filters were applied to retain 90% of the sequences in at least one sample for the non-dino community, and the dinoflagellate dataset. The *tax_glom()* function in the phyloseq package was utilized to aggregate the relative abundances of ASVs for assessing differences in composition at the class and genus levels. The R package ggplot2 (*Wickham, 2016*) was used to create graphs.

After data preprocessing and aggregation at the genus level, genera within both the non-dino community and dinoflagellate community were categorized into three trophic groups (phytoplankton, protozooplankton, mixoplankton) based on the work of (*Schneider et al., 2020*). These genera were also classified into two groups according to their lifestyle: parasitic and non-parasitic. We excluded genera where bootstrap support was below 80%. For taxonomic groups not annotated at the genus level, we used the immediately higher taxonomic category with a confirmed assignment. For genera with representatives in more than one trophic or lifestyle group, taxonomy at the species level was used, although this level was not always automatically annotated. To assign the taxonomy at the species level in these cases, ASVs were annotated with the highest score against NCBI 18S rRNA and nt/nr databases (16-03-2024), using BLASTN 2.15.0+ (*Altschul et al., 1990*), provided they agreed with the PR[2] taxonomy initially assigned. The trophic and lifestyle groups categorization was complemented with annotations of species/genus-specific information from other studies (*Gran-Stadniczeňko et al., 2019*; *Xiong et al., 2021*; *Genitsaris et al., 2022*; *Lapeyra Martin et al., 2022*).

To visualize and count the most abundant ASVs from the non-dino and dinoflagellate communities that are shared or exclusive between sponge species and SW, we created UpSet graphs using the ComplexUpSet package (*Lex et al., 2014*; *Krassowski, 2020*). A heatmap showing the relative abundance of dominant dinoflagellate ASVs across all samples was generated using the *plot_heatmap()* function in phyloseq.

Dino-Group-1-Clade 1 ASVs were placed in the reference phylogenetic tree using the RAxML evolutionary placement algorithm (*Stamatakis, 2014*). The reference tree was constructed from full-length 18S rRNA gene sequences of Dinoflagellata with the Apicomplexa sequence as the outgroup. The sequences were aligned using MAFFT v7.515 (*Katoh et al., 2002*), and the phylogenetic tree was constructed using RAxML v8.2.12 with

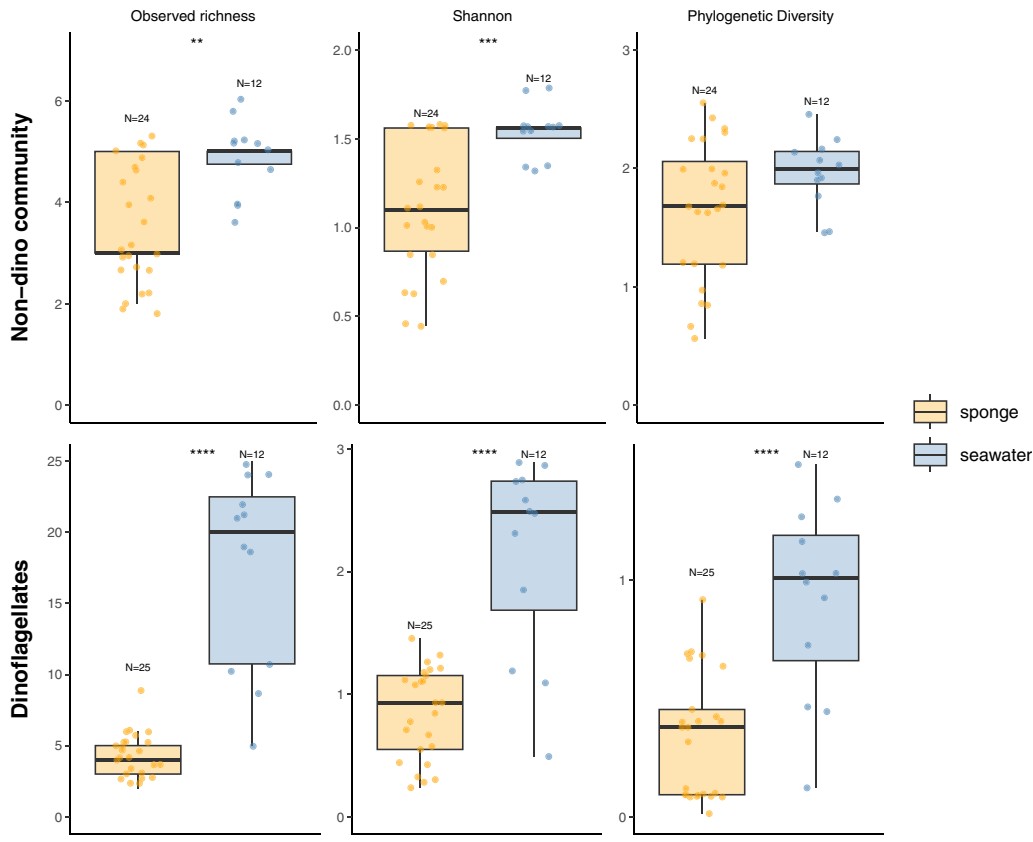

**Figure 1 Alpha diversity indexes (Observed richness, Shannon, and Phylogenetic diversity) of eukaryotic microbial communities associated with Antarctic sponges (orange) and SW (blue).** The analysis includes the non-dino community (upper panel) and dinoflagellates (lower panel). Asterisks indicate statistically significant differences determined by the Wilcoxon rank-sum test: **$p$ = 0.0013, ***$p$ = 0.0008 and ****$p$ = 4.213e-06, 5.591e–05 and 8.065e–05. Non-significant comparisons were omitted. Dots represent individual samples, and N denotes the number of samples in each group. Indexes computed with rarefied data to the minimum sampling depth (*Iophon* sp. and *Myxilla* (*Burtonanchora*) sp._1_2020 for non-dino community and dinoflagellates, respectively).

the General Time Reversible Categorical (GTRCAT) model. The resulting tree was modified for visualization with FigTree v1.4.4 (*Rambaut, 2018*). Raw sequencing data for this study was deposited on SRA under PRJNA1069634 and PRJNA1069605 for sponge microbiome and SW, respectively.

## RESULTS

### Diversity, community composition, trophic modes, and lifestyle of microbial eukaryotes, especially dinoflagellates associated with Antarctic sponges

We analyze the diversity and community composition of microbial eukaryotes associated with Antarctic sponges. For this purpose, we sampled 26 sponges from the Demospongiae class, including at least eight species from two orders and six families. We also analyzed 12

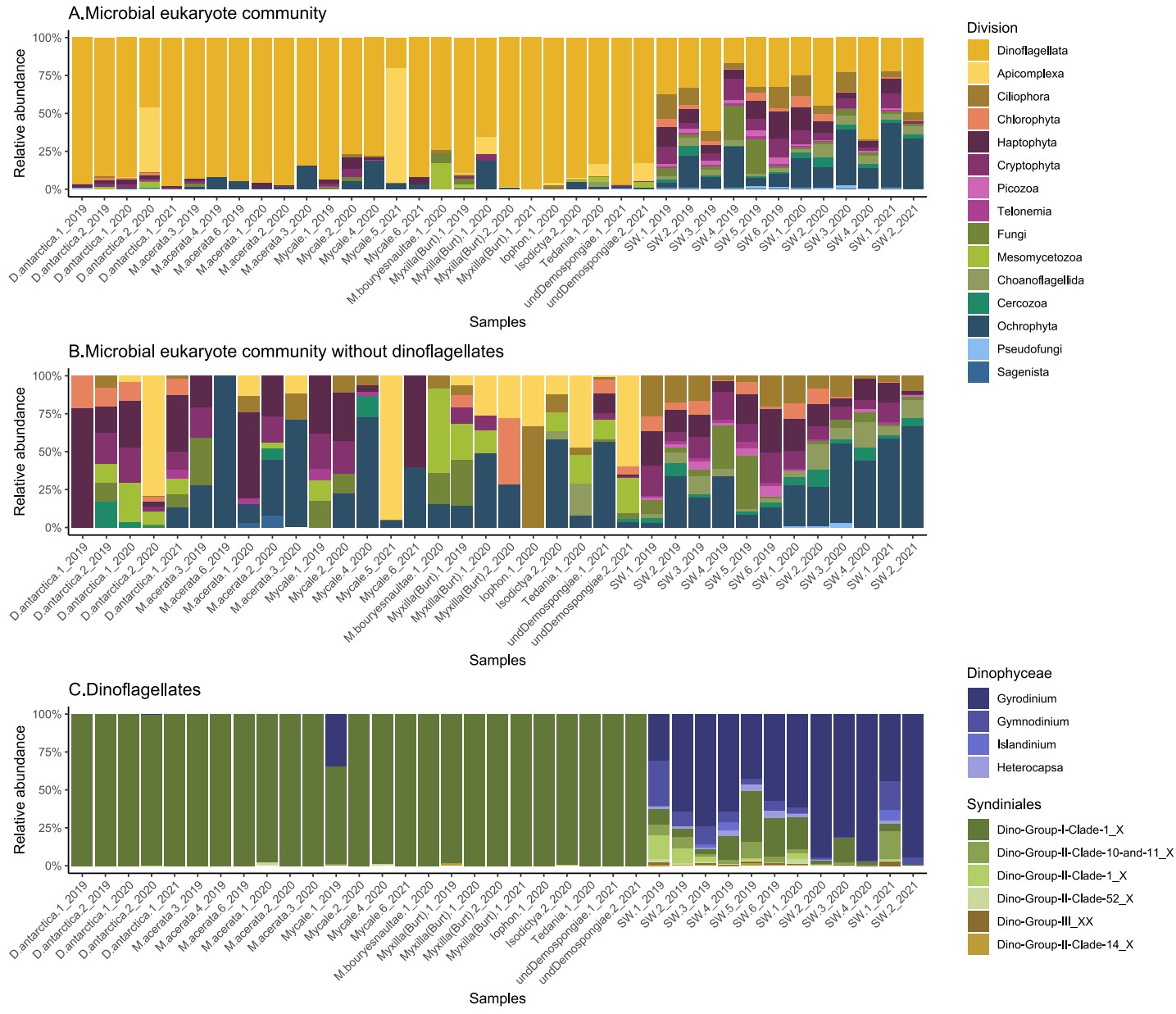

**Figure 2** **Composition of microbial eukaryotes in Antarctic sponges and SW.** Taxonomic composition at the division level of the microbial eukaryote community, encompassing dinoflagellate and non-dino communities (A) and microbial eukaryotes without dinoflagellates (B). (C) Dinoflagellate taxonomic composition at the genus level. Classes grouped the dinoflagellate genera. Relative abundances were calculated, with normalization by median sequencing depth. Sample names are denoted using the sponge species, replicate and summer of sampling. D. antarctica: *Dendrilla antarctica*; M. acerata: *Mycale acerata*; M. bouryesnaultae: *Mycale bouryesnaultae*; Myxilla (Burt): *Myxilla (Burtonanchora)* sp.; unde-Demospongiae: undetermined Demospongiae; SW: surrounding seawater.

seawater (SW) samples collected from Chile Bay (WAP) during the austral summers of 2019, 2020, and 2021 (Table S1).

We recovered 398 ASVs and 889,127 sequences for the general microbial eukaryote community (Table S2). In the microbial eukaryote community without dinoflagellates (non-dino community), we obtained 438,458 sequences and 283 ASVs. Regarding the

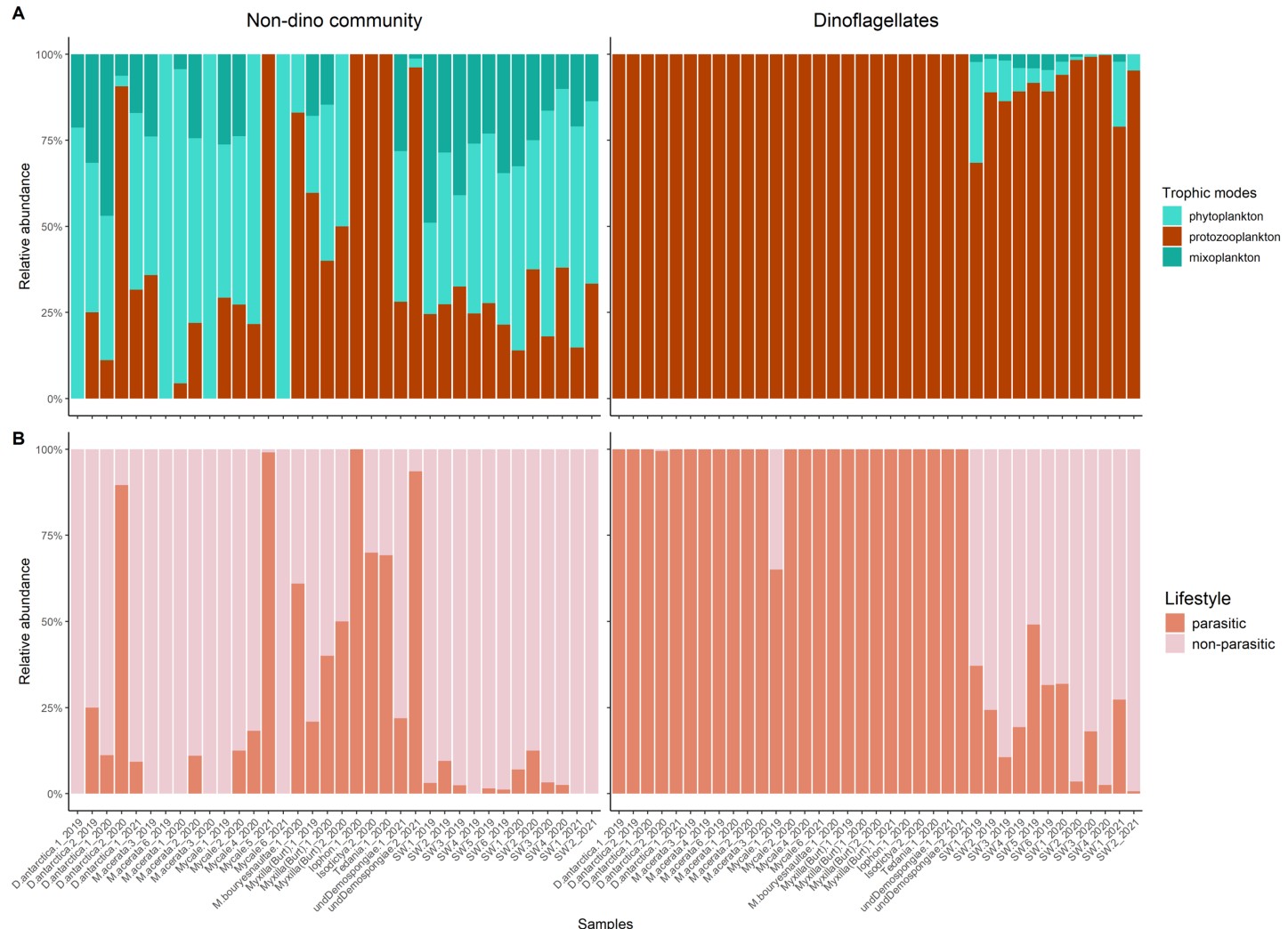

**Figure 3 Trophic modes and lifestyles of eukaryotic microbes associated with Antarctic sponges and surrounding SW.** (A) Barplots show the relative abundance of trophic modes within the non-dino (left panel) and dinoflagellates communities (right panel). (B) Relative proportions of lifestyles (parasitic or non-parasitic) in the non-dino community (left panel) and dinoflagellates (right panel). Trophic modes and lifestyles were annotated based on taxonomic assignments at the genus level of ASV, representing the top 90% of most abundant sequences in at least one sample. Sample names are denoted using the sponge species, replicate and summer of sampling. *D. antarctica: Dendrilla antarctica*; M. acerata: *Mycale acerata*; M. bouryesnaultae: *Mycale bouryesnaultae*; Myxilla (Burt): *Myxilla (Burtonanchora)* sp.; undeDemospongiae: undetermined Demospongiae; SW: surrounding seawater.

dinoflagellates, we obtained 450,669 sequences grouped into 115 ASVs. The alpha diversity analysis revealed significantly lower richness and diversity when comparing sponges to the SW for the non-dino and dinoflagellate communities (Fig. 1, Table S3).

The general community composition of microbial eukaryotes associated with Antarctic sponges shows a clear predominance of Dinoflagellata (85 ± 18%), while other groups were also notable in the SW, like Ochrophyta and Haptophyta (Fig. 2A). However, when we visualize the non-dino community, other important divisions appear, such as Apicomplexa, and Mesomycetozoa (Fig. 2B). At the class level, the most abundant microbial eukaryotes from the non-dino community were Bacillariophyta (Ochrophyta)

(27 ± 26%) and Prymnesiophyceae (Haptophyta) (16 ± 19%). At the same time, Coccidiomorphea, Gregarinomorphea (Apicomplexa), and Ichthyosporea (Mesomycetozoa) were observed only in sponges (Fig. S1).

Protozooplankton and phytoplankton were the dominant groups in sponges (Fig. 3A). We detected differences in trophic modes among sponge genera within the non-dino community. Specifically, phytoplankton was most abundant in *Dendrilla* and *Mycale*, whereas protozooplankton predominated in *Myxilla* (*Burtonanchora*) (Fig. 3A). In the dinoflagellate communities, protozooplankton dominated all sponge genera (Fig. 3A). Dinoflagellates were mainly represented by parasites, whereas non-parasitic microorganisms dominated in the non-dino community (Fig. 3B, Tables S4, and S5). Within the non-dinoflagellate community, *Rhytidocystis* (Coccidiomorphea) and *Paralecudina* (Gregarinomorphea) were the most abundant parasitic genera (7 ± 22% and 9 ± 22%, respectively) (Table S4).

The nMDS based on Aitchison distance revealed a clear clustering pattern by habitat (sponges and SW) for non-dino and dinoflagellate communities (Fig. S2). ANOSIM confirmed that the composition of the non-dino and dinoflagellate communities was significantly different ($p = 0.001$ Table S6).

When focusing on high-abundance non-dino microbial eukaryotes, ASV102 (*Paralecudina*), exclusive from sponges and belonging to the class Gregarinomorphea, was the most shared, found in at least five species and representing 3.43% of the total abundance. On the other hand, ASV080 (*Rhytidocystis*) from the class Coccidiomorphea was identified in *Dendrilla antarctica*, an undetermined Demospongiae, and *Mycale* sp., with the highest proportion in the latter (85%) and absent in SW (Fig. S3A). Considering only the most predominant dinoflagellate ASVs, ASV005, belonging to the Syndiniales class, was shared across all sponge species (Fig. S3B). Together with ASV016 and ASV023, ASV005 were highly abundant in sponges but low in SW (96% *vs.* 11%) (Fig. S4).

## Interannual changes in the diversity and community composition of microbial eukaryotes, especially dinoflagellates, associated with Antarctic sponges

Overall, the sponge-associated dinoflagellate community was more stable than the SW community due to the almost complete dominance of Syndiniales Dino-Group-I-Clade-1 in the former (98 ± 7% *vs.* 12 ± 11%) (Fig. 2C).

For the three genera of sponges present in the 3 years (*Dendrilla*, *Mycale*, and *Myxilla* (*Burtonanchora*)) pairwise comparisons between summers indicated significant differences in Shannon-Wiener alpha diversity index only for the dinoflagellate communities (Fig. 4). Dinoflagellates and non-dino communities associated with Antarctic sponges show compositional homogeneity between the different summers (Fig. 5A), with the dinoflagellate community associated with *Mycale acerata* distinct from other species. PERMANOVA analysis at the ASV level revealed significant variations among host genera (non-dino community $p = 0.024$; dinoflagellates $p = 0.006$, Table S6). However, a dispersion test using PERMDISP revealed significant differences in heterogeneity ($p = 0.04$) among host genera for the non-dino community. Interestingly, the distribution

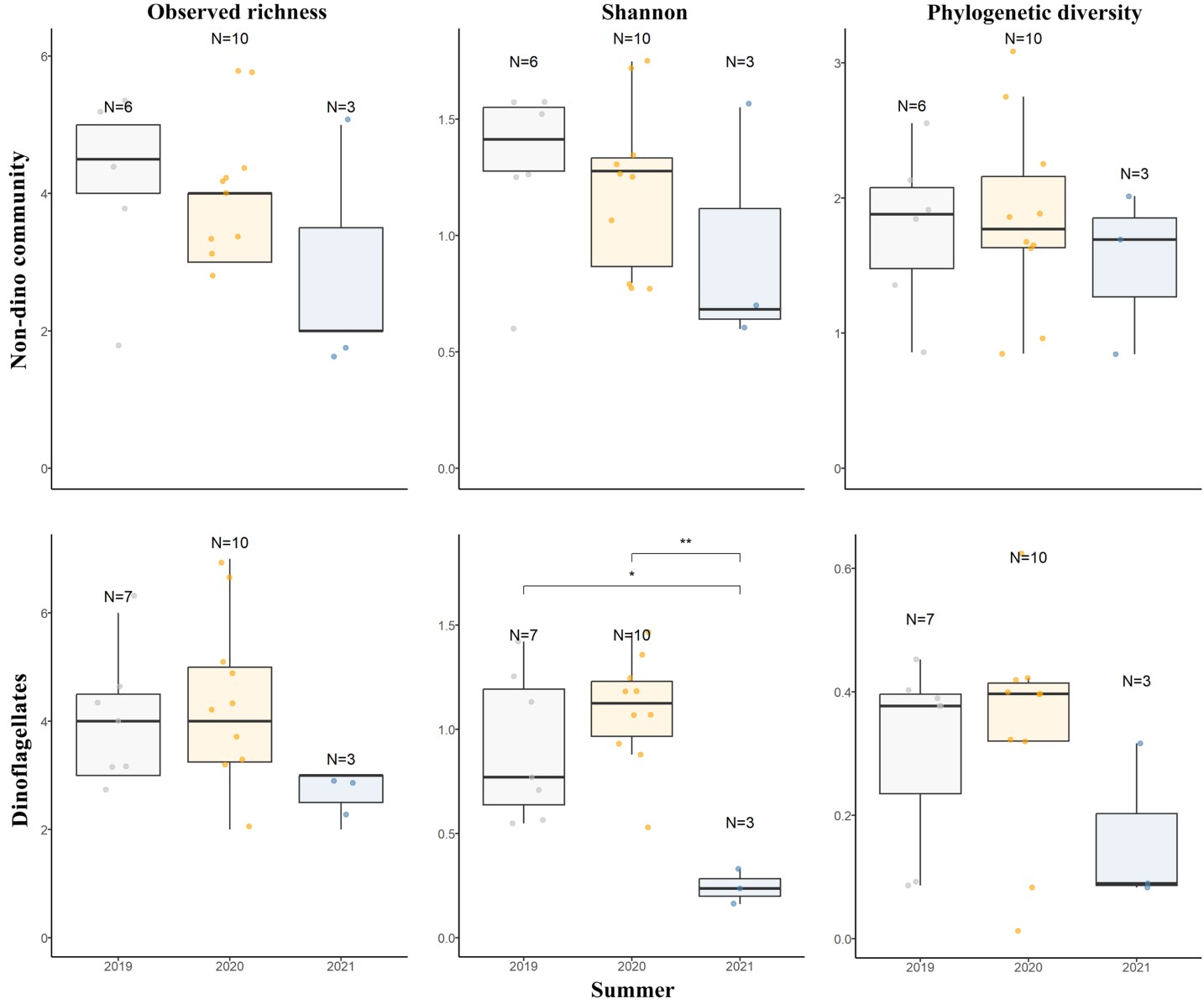

**Figure 4 Diversity measures of eukaryotic microbial communities associated with Antarctic sponges across different summers.** Observed richness, Shannon, and Phylogenetic diversity ecological indexes of the non-dino community (upper panel) and dinoflagellates (lower panel) associated with Antarctic sponges over three summers. Significant differences in Shannon diversity across summers were assessed using the Kruskal-Wallis test ($\chi^2$ = 8.1253, df = 2, $p$ = 0.01976). Asterisks denote levels of statistical significance as established by the Wilcoxon rank-sum test: *for $p$ = 0.017 and **for $p$ = 0.007. Comparisons without significance are not shown. Points indicate separate samples, while N signifies the sample count per group. The analysis represents sponge genera resent across the three summers (*Dendrilla*, *Mycale*, and *Myxilla* (*Burtonanchora*)).

of ASVs belonging to the Syndiniales Dino-Group-I-Clade-1 (ASV005, ASV016, and ASV023) across *Dendrilla*, *Mycale*, and *Myxilla* over the years indicates a consistent pattern, typically comprising >75% of the relative abundances (Fig. 5B). Additionally, we detected a noticeable increase in the relative abundance of ASV_005 between years, moving from 35 ± 38% in 2019 to 94 ± 5% in 2021. Dino-Group-I-Clade-1 ASV005 was highly abundant in sponge samples, especially within a 2021 *Mycale* sp. sponge, where it

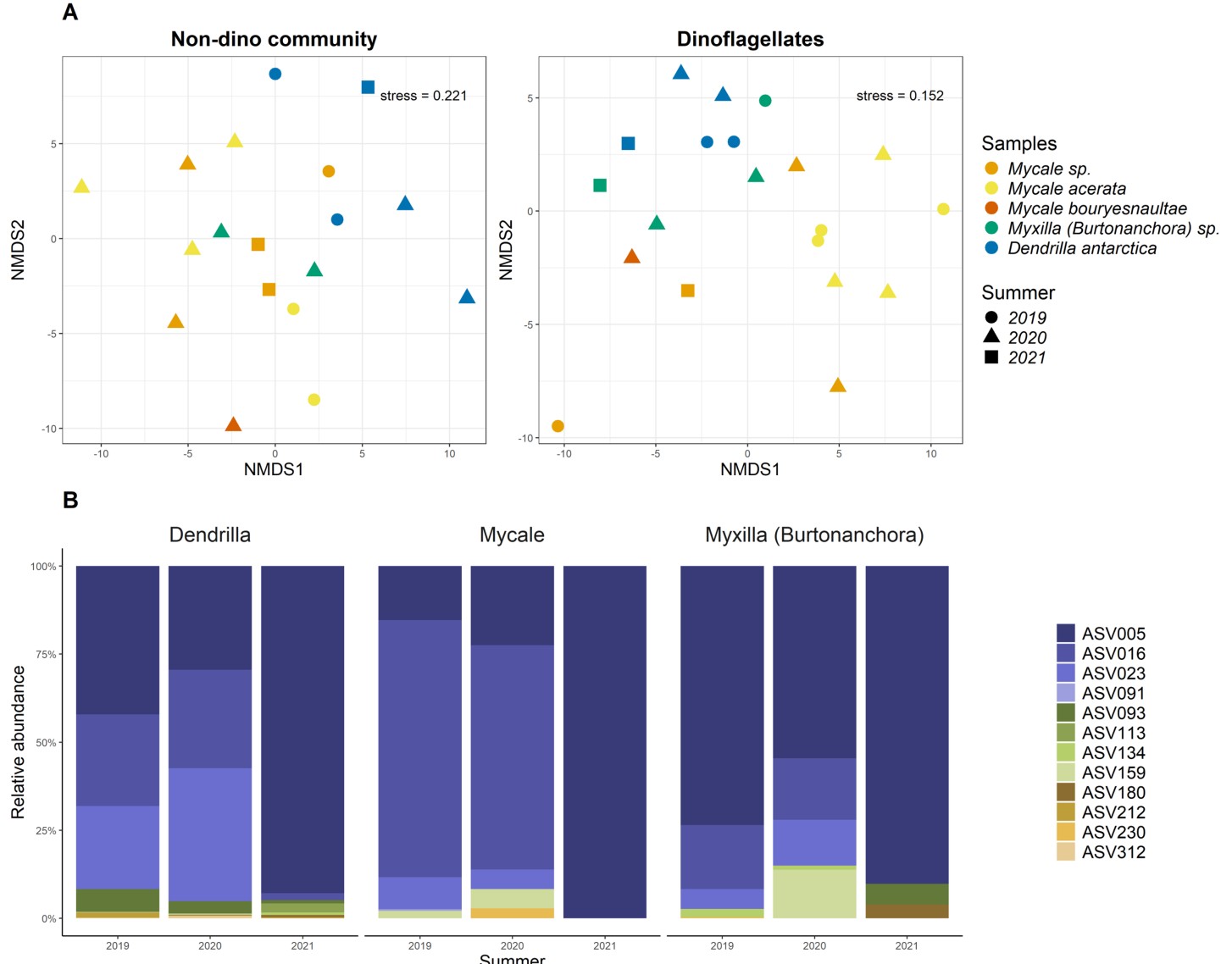

**Figure 5  Distributions of microbial eukaryotes associated with Antarctic sponges over three summers and interannual dynamics ASVs from Dino-Group-I-Clade 1.** (A) Non-metric Multidimensional Scaling (nMDS) plots for the non-dino community (left panel) and dinoflagellates (right panel) associated with Antarctic sponges, based on ASV-level and using Aitchison distances. The ordination used centered log-ratio-transformed relative abundances of microbial taxa. PERMANOVA analysis with 999 iterations revealed significant differences between summers within the dinoflagellate community ($p = 0.103$), unlike the non-dino community ($p = 0.373$). ASVs with an abundance lower than 0.1% were removed. (B) Distribution and relative abundance of ASVs from Dino-Group-I-Clade 1 across summers. The analyses are restricted to sponge genera with replicates over three summers.

comprised virtually 100% of dinoflagellate reads (Fig. 5B). The Dino-Group-I-Clade-1 ASV005 is the most abundant dinoflagellate in our dataset and was the only one shared between all sponge species analyzed (Fig. S3B). Phylogenetic analysis of the Dino-Group-I-Clade-1 clade ASVs (ASV005, ASV016, and ASV023) revealed their relationship with uncultured Syndiniales from temperate waters of the Mediterranean (Fig. S5).

## DISCUSSION

Here, we used 18S rRNA gene sequencing to investigate the diversity, community composition, trophic strategies, and lifestyle of microbial eukaryotes associated with Antarctic sponges over 3 years, primarily focusing on the dinoflagellate community. A large number of individuals and a high richness of the sponge species used in this study enabled a detailed resolution of the dynamics of the microbial eukaryotes associated with Antarctic sponges.

Trophic modes of plankton refer to their methods of carbon acquisition and have been traditionally categorized into autotrophy, where phytoplankton use photosynthesis to convert carbon dioxide and sunlight into organic matter, forming the foundation of the aquatic food web, and heterotrophy, where zooplankton and bacterioplankton consume organic material, including other plankton, detritus, and dissolved organic matter, thus recycling nutrients within the ecosystem (*Glibert & Mitra, 2022*). Mixotrophy represents a flexible strategy where organisms can switch between autotrophy and heterotrophy depending on environmental conditions, allowing them to optimize their nutritional intake (*Mitra & Flynn, 2010*; *Mitra et al., 2014*). Recent studies have revealed that mixotrophy is much more common and widely distributed among plankton than previously known, highlighting its significant role in marine ecosystems (*Flynn et al., 2019*; *Mitra et al., 2023*; *Millette et al., 2023*). In our study, trophic analysis was performed based on a database of trophic modes of aquatic protists by *Schneider et al. (2020)* and detailed curation of the assigned taxonomy. Our results showed that almost all sponge-associated dinoflagellates were annotated as protozooplankton and most of them were classified as parasites. These results underscore the complex trophic interactions and specialized ecological roles dinoflagellates play in the Antarctic sponge microbiome, contributing to the stability and functioning of these unique marine ecosystems. Understanding the trophic strategies is crucial for insights into how parasitic dinoflagellates contribute to the stability and ecological dynamics of Antarctic sponge communities. Studying molecular taxonomy, ecology, and physiology of microbial eukaryotes and specifically dinoflagellates can be particularly challenging. Dinoflagellates represent a highly diverse taxonomic group with a high degree of horizontal gene transfer (*Wisecaver, Brosnahan & Hackett, 2013*) and a high number of 18S rRNA copies within a single cell (*Gong & Marchetti, 2019*). Additionally, they possess large and complex genomes and exhibit nutritionally versatile and complex life histories (*Murray et al., 2016*). Moreover, the environmental diversity of dinoflagellates currently needs to be better reflected in culture collections or sequenced genomes, which hinders the proper description of species and, consequently, the availability of suitable reference sequences for taxonomic assignment. In the present study, many taxa could not be taxonomically assigned to the genus level, including dominant groups, which unfortunately limits a fine-scale understanding of the structure of microbial eukaryote and dinoflagellate communities.

## Dinoflagellate communities associated with Antarctic sponges are distinct from seawater and remain stable over time

The dinoflagellate community associated with sponges differed from its SW counterparts in diversity and community composition. Recent studies have highlighted the importance of dinoflagellates as abundant microeukaryotes within Antarctic sponges (*Rodríguez-Marconi et al., 2015*; *Moreno-Pino et al., 2024*). However, while previous studies utilized the V9 region (*Rodríguez-Marconi et al., 2015*; *Moreno-Pino et al., 2024*), we used the V4 region. The V9 region captures more diversity and is better suited for amplifying genotypes, especially those of low abundance. On the other side, the V4 region is more appropriate for higher taxonomic resolution because its high divergence and sequence variation offer more significant species-level taxonomic distinctions of dinoflagellates (*Stoeck et al., 2010*; *Ki, 2012*).

Our results revealed interannual stability in dinoflagellate communities associated with Antarctic sponges. However, we observed differences in the Shannon index when analyzing the impact of summers. This apparent contradiction could be attributed to the limited number of sponge samples collected in 2021, potentially leading to an underestimation of the richness of the dinoflagellate community. Smaller sample sizes generally yield fewer detected species due to chance (*Roswell, Dushoff & Winfree, 2021*), directly impacting richness measurement and influencing the Shannon index, which emphasizes species richness (*Kim et al., 2017*). Other explanation could be the dominance of ASV005 in the sponge samples collected in 2021, which could have led to an uneven distribution of species within the dinoflagellate community. In this point, it is important to note that previous studies documented the temporal stability of the bacteriome in Antarctic sponges, even in the face of sudden increases in SW temperature (*Cárdenas et al., 2019*; *Happel et al., 2022*). On the other hand, in the sponge *Rhopaloeides odorabile* from the Great Barrier Reef, the Bacteria, Archaea, and Eukarya microbial communities also remained stable across multiple nutrient and temperature stressors (*Simister et al., 2012*). In this way, the stability of the microeukaryotic communities, mainly the dinoflagellates associated with our Antarctic sponges, could suggest they are advantageous for the host in this extreme and changing ecosystem.

## Protist parasites in Antarctic sponges

Apicomplexa, Mesomycetozoa, and parasite groups from dinoflagellates dominate the microeukaryotic community of Antarctic sponges. The Apicomplexa phylum encompasses intracellular symbionts in Antarctic, temperate, and tropical sponges; but the nature of their interactions has not been elucidated (*Moreno-Pino et al., 2024*; *Chaib De Mares et al., 2017*; *Cleary, 2019*; *Ferreira & Cleary, 2022*). Although typically classified as obligate parasites, they can interact with their hosts, including mutualism and commensalism, as seen in coral microbiomes with corallicolids (*Saffo et al., 2010*; *Kwong et al., 2019*). The genus *Rhytidocystis* was highlighted as a major taxon of Coccidiomorphea (Apicomplexa, ASV080), although only in three of the 26 sponge microeukaryomes analyzed. It is known as a coccidian-like parasite of polychaetes but can also be found in mollusks, flatworms, and arthropods (*Rueckert & Leander, 2009*; *Holt et al., 2022*). To our

knowledge, there is no evidence that *Rhytidocystis* has been described in Antarctica either as a sponge or coral symbiont. However, sponges are recognized for providing microhabitats for polychaetes and other invertebrates, particularly some associated with Antarctic sponges that spend most of their life cycle within these hosts (*McClintock et al., 2005*). In this way, the presence of *Rhytidocystis* in Antarctic sponges could be linked to its host polychaetes residing within the sponge tissues.

We also detected *Paralecudina* (class Gregarinomorphea, order Eugregarinorida, Apicomplexa, ASV102), reported previously in Antarctica as a parasite of krill (*Cleary, 2019*). Eugregarines are host-specific parasites found in the intestines, coeloms, and reproductive vesicles of marine invertebrates (*Leander, 2008*; *Rueckert, Betts & Tsaousis, 2019*). Symbiotic relationships with gregarines range from antagonistic to beneficial effects (*Rueckert, Betts & Tsaousis, 2019*). Eugregarines are widespread in tropical and temperate sponges and play a significant role in marine food web dynamics by regulating host populations and influencing the carbon cycle (*Chaib De Mares et al., 2017*; *Del Campo et al., 2019*). In Antarctica, eugregarines dominate the sediment and are among the most abundant orders within the sponge microeukaryome, suggesting that the sediment could serve as a microbial seed bank for sponges (*Cleary & Durbin, 2016*; *Moreno-Pino et al., 2024*).

Ichthyosporea (Mesomycetozoa) were previously discovered in the water column and sediment of the WAP (*Cleary & Durbin, 2016*). However, they had now been identified in Antarctic sponges. Ichthyosporea is a class that includes osmotrophic microorganisms found in fish, amphibians, and bivalves. Some of these species are mutualists that reside in the digestive tracts of invertebrates, while others are parasites that can lead to the death of their hosts (*Mendoza, Taylor & Ajello, 2002*; *Glockling, Marshall & Gleason, 2013*). Previous studies have found Mesomycetozoa in only one sponge at very low abundance in Fildes Bay, Antarctica (*Moreno-Pino et al., 2024*), and at abundances less than 1% in sponges from the Caribbean and Mediterranean (*Chaib De Mares et al., 2017*), in contrast to this study, where abundances reached up to 55%. Thus, it is likely that the differences in abundance result from primer bias. The selection of gene regions, primers, and databases can significantly affect microbial community diversity and identification (*Stuart et al., 2024*). We evaluated the best primers for detection of Apicomplexa, Dinoflagellata, Stramenopiles, Haptophyta, and Ichthyosporea, finding they amplify more than 75% of sequences within the PR2 database for most groups (https://app.pr2-primers.org/pr2-primers/; *Vaulot et al., 2022*). In particular, the primers used in this study have a higher resolution for Ichthyosporea than previous studies, amplifying 90% of sequences from this class within the PR2 database (https://app.pr2-primers.org/pr2-primers/; *Vaulot et al., 2022*).

Our previous analysis indicate that only the primers in *Cleary et al. (2019)* and *Ferreira & Cleary (2022)* offer resolutions similar to this study. In contrast, those in *Moreno-Pino et al. (2024)*, *Chaib De Mares et al. (2017)* and *Hardoim et al. (2021)* amplify less than 25% of the most abundant groups detected here. A common limitation in microeukaryote analyses is the need for many databases to resolve taxa at the species level fully (*Del Campo et al., 2018*). The PR2 database, chosen for its extensive coverage and accuracy for marine

microbial eukaryotes (*Guillou et al., 2012*), identifies more OTU and resolves taxa to deeper levels than the SILVA database (*Kataoka & Kondo, 2019*).

## Dominance of Syndiniales Dino-Group-I-Clade 1 in Antarctic sponges

The most abundant parasite group from dinoflagellates detected here is the Syndiniales Dino-Group-I-Clade-1, an important player in ice-edge environments (*Clarke et al., 2019*). Syndiniales, a type of marine alveolates, are present almost everywhere and can infect and kill a wide range of hosts, including other protists (such as dinoflagellates, ciliates, and radiolarians) and metazoans (such as copepods, cladocerans, and polychaete larvae) (*Guillou et al., 2008*). Syndiniales have been reported to be abundant in sponges from temperate and tropical waters (*Hardoim et al., 2021*; *Chaib De Mares et al., 2017*), suggesting differences in thermal tolerances or distribution patterns among their strains and species (*Anderson & Harvey, 2020*). Specifically, Dino-Group-I-Clade-1 has been found in high abundance in tropical sponges (*Cleary, 2019*; *Ferreira & Cleary, 2022*). Together, these results demonstrate that parasites are abundant in Antarctic sponges and could significantly influence nutrient dynamics and the community of microorganisms associated with them; however, further research is needed to precisely define these interactions and the ecological roles of these parasites.

The ASVs assigned to Syndiniales Dino-Group-I-Clade 1, ASV005, ASV016, and ASV023 were consistently dominant in sponge tissues and very low in the surrounding SW. This indicates they correspond to sponge-enriched sequence clusters (SESC). Previous studies have reported the SESC of bacteria and fungi (*Webster et al., 2010*; *Nguyen & Thomas, 2018*). These findings suggest that these taxa, enriched by the host sponge's filtration, may have advantages and adapt to the ecological niches within sponges. The enrichment of ASV0016 in the sponge *Mycale acerata* reveals a unique, distinctive dinoflagellate community and suggests a specific selection. Additionally, the differences in the relative abundances of the same ASVs across sponges emphasize the preferential of Syndiniales for certain sponge species.

We identified three dominant and recurrent Syndiniales Dino-Group-I-Clade 1 ASVs in the summers. Among these, ASV005 was found to be the most dominant in the year 2021. The recurrence of Syndiniales over time has been significantly associated with the host's biomass, density, and cyclical patterns, such as annual blooms (*Rizos et al., 2023*; *Anderson & Harvey, 2020*). Notably, similar positive associations between Dino-Group I and ciliates, apicomplexans, have also been observed in coral microbiomes (*Bonacolta et al., 2024*).

The marine ecosystem has undergone significant changes due to the acceleration of glacial melting, leading to the rapid warming of the climate and the cooling of Southern Ocean waters. These changes have resulted in modifications to the dominant phytoplankton species, with large diatoms being replaced by cryptophytes and small flagellates (*Moline et al., 2004*; *Queirós et al., 2024*). In this changing environment, Syndiniales parasites have emerged as critical players in the structure of the oceanic food web (*Cleary & Durbin, 2016*). The dominance of Syndiniales Dino Group I has been previously observed in tropical and temperate sponges (*Chaib De Mares et al., 2017*;

*Cleary, 2019*; *Ferreira & Cleary, 2022*), suggesting a crucial ecological interaction. Parasite-rich ecosystems benefit biodiversity by modulating host population dynamics and influencing energy flow (*Hudson, Dobson & Lafferty, 2006*). In Antarctic marine ecosystems, high-specificity parasites may help maintain diversity and shape seasonal succession by reducing the fitness or causing the mortality of abundant species, thereby creating opportunities for other species within the ecosystem (*Cleary & Durbin, 2016*).

## CONCLUSION

This study expands our understanding of the microeukaryome of sponges in polar regions, and is the first to provide a detailed characterization of dinoflagellates associated with Antarctic sponges. Our findings indicate that dinoflagellates dominate the microbial eukaryote communities associated with Antarctic sponges. We also observed no interannual variations in the microeukaryotic communities associated with Antarctic sponges, and that parasitic dinoflagellates are remarkably stable and dominant, suggesting that parasites in the Antarctic marine environment are crucial for ecosystem engineers as sponges. Syndiniales, specifically Dino-Group-I clade 1, with their ability to regulate host populations, may play a crucial role in the microbiome of Antarctic sponges. Future research should further explore the roles of these parasitic dinoflagellates within Antarctic sponges, which can include carbon fluxes and responses to ongoing environmental changes in these primitive hosts.

## ACKNOWLEDGEMENTS

We are grateful to Dr. Fernando Alfaro, Dr. Mario Moreno-Pino, Dr. Leslie Daille, Dr. Gustavo Rodriguez, and Dr. (c) Patricio Flores-Herrera for their suggestions and comments during the data analysis. Many thanks also to Marlene Manzano for assistance in the laboratory. Additionally, we thank Andreas Schmider, Lea Happel, Yann Herrera, José Hernández, Alberto Ahumada, and Vicente Villalobos for sponge sampling.

### Funding

This research was supported by Agencia Nacional de Investigación y Desarrollo (ANID), Fondecyt Grant N° 1230758, Subdirección de CapitalHumano/Doctorado Nacional/2019-doctoral fellowship N° 21192150 and INACH DG_15-20 Grant. The funders had no role in study design, data collection and analysis, decision to publish, or preparation of the manuscript.

### Grant Disclosures

The following grant information was disclosed by the authors:
Agencia Nacional de Investigación y Desarrollo (ANID) Fondecyt: N° 1230758.
Subdirección de CapitalHumano/Doctorado Nacional/2019-doctoral fellowship: N° 21192150 and INACH DG_15-20.

## Competing Interests

The authors declare that they have no competing interests.

## Author Contributions

- Marileyxis R. López-Rodríguez conceived and designed the experiments, performed the experiments, analyzed the data, prepared figures and/or tables, authored or reviewed drafts of the article, and approved the final draft.
- Catherine Gérikas Ribeiro analyzed the data, prepared figures and/or tables, authored or reviewed drafts of the article, and approved the final draft.
- Susana Rodríguez-Marconi performed the experiments, authored or reviewed drafts of the article, and approved the final draft.
- Génesis Parada-Pozo performed the experiments, authored or reviewed drafts of the article, and approved the final draft.
- Maria Manrique-de-la-Cuba performed the experiments, authored or reviewed drafts of the article, and approved the final draft.
- Nicole Trefault conceived and designed the experiments, analyzed the data, authored or reviewed drafts of the article, and approved the final draft.

## Field Study Permissions

The following information was supplied relating to field study approvals (*i.e.*, approving body and any reference numbers):

Sampling was performed under permits from the Chilean Antarctic Institute (INACH).

## Data Availability

The 18S raw sequence reads are available at SRA: PRJNA1069634.

## Supplemental Information

Supplemental information for this article can be found online at http://dx.doi.org/10.7717/peerj.18365#supplemental-information.

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
