# Peer review of "Stable dominance of parasitic dinoflagellates in Antarctic sponges"

_PeerJ, doi:10.7717/peerj.18365_

## Round 0.1 · original submission · Major Revisions

The reviewers clearly see merit in your study and consider it of interest for the broad readership of PeerJ, an assessment with which I wholeheartedly agree.

The most pressing concerns and points raised by the reviewers include:

• Clarifications regarding the sponge sampling strategy and region of choice for the 18S rRNA gene for sequencing as requested by reviewer 2;
• Including a discussion on the results of the trophic analysis performed in your work,
• Inclusion of supplementary sequencing data (unfiltered ASV and ASV sequence fasta files),
• Reviewer 1 specifically suggests additional analyses for characterization of a) beta diversity and for b) elucidation of evolutionary and ecological context of parasitic dinoflagellate taxa to better support your findings about host specificity (for details, pls refer to the reviewers’ report below).
• Both reviewers strongly recommend making all code publicly available (e.g. deposition of codes in a repository, such as e.g. github).

Thank you for taking the time to revise your manuscript. I am looking forward to receiving a revised version.

·

Basic reporting

Overall, the findings of this manuscript are clearly reported and represent a unique and interesting study relevant to those researching marine microbiomes, sponge ecology, and marine parasites. I suggest some minor reporting edits below in order to convey the results of this interesting study in a more effective way.

Line 49- Some information here on the different ecological roles of Antarctic vs tropical sponges would be great to provide additional context for how unique Antarctic communities can be.

Line 53- I would change “named” here to “referred to as a”

Line 82- Has the temporal patterns of the microeukaryome been studied thoroughly in other marine hosts? If so, a quick description of those findings and how they guide your hypothesis would go well here? If not, then you should further highlight the novel nature of this study!

Lines 82-84- I would rephrase this sentence if possible in order to avoid hyperbole such as “least known” and “most exciting”. While I agree that not much is known about them in sponges and that this is exciting to explore, I think a more objective statement here would suffice.

Overall, more details in the introduction in regards to why Dinoflagellates were singled out and targeted so that the average reader understands the reasoning for this research would be useful to include here.

Line 211- Missing the ggplot2 citation

Line 381- I would remove “fungi-like” here in order to avoid confusion.

Line 394- I would replace “important” here with a more objective term.

The discussion highlights the hypothesized contributions of specific microbes well, however a broader discussion focusing on the trophic analysis that was performed would do well here.

Figure 2- This figure can be improved by better marking the different species of sponges and the different sampling years. The sample names (specifically the _E** portion), while useful and understood by the authors, are not easily understood by the reader.

Experimental design

I see Bray-Curtis was used for the beta-diversity analyses. While this method is commonly reported in microbiome studies and I do feel like its inclusion is overall helpful to the analysis, the field is shifting towards Aitchison distances for beta-diversity analyses as this approach better accounts for the compositional nature of metabarcoding data according to Gloor et al 2017 (https://www.ncbi.nlm.nih.gov/pmc/articles/PMC5695134/). I therefore recommend adding this to the analysis.

I commend the authors for their thorough and robust approach in regard to assigning taxonomy to their ASVs.

Please include the unfiltered ASV table and ASV fasta sequences (at least for the sequences noted in the supplementary tables).

Please also provide the code used for the analysis through an online platform such as github or as supplementary material if possible.

Validity of the findings

Considering the strong abundance of Dino-Group-1-Clade 1 and the relatively few ASVs it encompasses, I strongly feel that more analysis can be done to explore the nature of these sequences. I suggest performing any of the following in order to get a better understanding of the evolutionary and ecological context of these ASVs to better support your findings about their specificity:
- A phylogenetic approach incorporating EPA-placed ASVs into a phylogeny of Dino-Group-1-Clade 1.
- BLAST these sequences with the NCBI database to see where other closely related sequences have been recovered.

Overall, the goal of this extra analysis would be to see if the ASVs recovered here are closely related to other sponge-associated Syndiniales or Antarctic ones?

Reviewer 2 ·

Basic reporting

The manuscript is clearly written and well structured. The context used to introduce the study is relevant and well articulated into the larger research field.

The following kinds of data were generated along with the paper, and these were found to be 'self-contained':
- Sponge-associated amplicon sequences for the 18S rRNA V4 region
- Seawater amplicon sequences for the 18S rRNA V4 region
- Custom scripts for analyses in R

I was able to verify that amplicon sequences were made available in the case of the 26 samples identified as host-associated and for the additional 12 samples reported in the study as originating from seawater.

Scripts for data analysis in R do not seem to be available. These scripts should also be made available along with a proper documentation.

Experimental design

The research conducted in this study falls within the aims and scope of the journal.
The research question is well defined and it is also relevant. In general, experimental design was clearly stated. However, some aspects regarding decisions on the kinds of sponge species sampled across years and the location of sampling were not thoroughly detailed in the text. Specific comments and concerns are stated in the Additional Comments section.

Validity of the findings

The results are valid and linked to the original research question. The conclusions are supported by the results obtained.

Additional comments

It is timely and definitely needed to have contributions such as this study, addressing eukaryotic microbial associates in sponges, particularly in Antarctic habitats, where their abundance suggests a primary role. I have a few observations as stated below:

L 15-16: The authors open their abstract stating that "Dinoflagellates are essential members of various ecosystems, but their role as parasites within Antarctic sponge holobionts still needs to be better understood." - I do not think this sentence is to be debated, but gives an impression that the study will actually address the role of parasites in the sponges, conducting some sort of functional analysis, but the focus of the study is clearly to address community composition and diversity of microbial eukaryotes.
L 118-119 + L 122-123: It is not clear what sampling strategy was followed. The number of samples collected, whether a minimum distance was kept, or if some species were prioritized for analysis. Upon closer inspection of the supplementary file Table S1, it appears as if the collection coordinates in most cases overlaps exactly. Does that mean that particular individuals from recurring species were followed along time? If so, the statistical analysis in its current form does not take into account this factor.
L 142-144: The reason for selecting the hypervariable region V4 of the 18S rRNA gene is not clearly stated. The Earth Microbiome Project target the V9 region, and benchmarking studies have shown that V9 best represents a wider taxonomic diversity when compared with V4, even when it comes to detecting rare species. If this study aimed to characterize broadly the community composition in these Antarctic sponges, what is the argument behind the choice of region? And, more importantly, how could this introduce a bias in the taxonomic composition you observe from these samples?
L 389-391: The authors add an interesting analysis on primer bias for Ichthyosporea within the PR2 database. Is there a similar bias with other prominent groups discussed in the paper? The database choice is actually not discussed, but it may be important to consider, particularly since comparisons are made with other studies that used different databases.

---

## Round 0.2 · accepted · Accept

The authors have thoroughly addressed all reviewer comments, and the manuscript is now in good shape for publication in PeerJ.

·

Basic reporting

The authors have addressed all my comments and have added the background information necessary to effectively convey their findings.

Experimental design

I appreciate the authors providing the scripts and additional information requested (ASV tables, Aitchsion Distances, RAXML trees, etc.) and I am satisfied with the revision. The github link is confirmed to be accessible.

Validity of the findings

The additional analysis is well done and I am satisfied with the revision.

Reviewer 2 ·

Basic reporting

The manuscript is clearly written and well structured. The context used to introduce the study is relevant and well articulated into the larger research field.

The following kinds of data were generated along with the paper, and these were found to be 'self-contained':
- Sponge-associated amplicon sequences for the 18S rRNA V4 region
- Seawater amplicon sequences for the 18S rRNA V4 region
- Custom scripts for analyses in R

I was able to verify that amplicon sequences were made available in the case of the 26 samples identified as host-associated and for the additional 12 samples reported in the study as originating from seawater.

Scripts for data analysis in R are now available after review in a repository in GitHub, this is also now referred to in the main text.

Experimental design

In addition to comments in the former review, additional details on experimental design were now clearly stated in the text. Clarifications provided on the sampling strategy across years make clear that the statistical analyses performed are adequate.

Validity of the findings

The results are valid and linked to the original research question. The conclusions are supported by the results obtained.

Additional comments

The authors provided satisfactory answers to the comments and suggestions given in the former round of review, and updated the manuscript accordingly.